# Marine Environmental Capacity in Sanmen Bay, China

Yanming Yao [1], Jiahao Zhu [1], Li Li [1,2,*], Jiachen Wang [1] and Jinxiong Yuan [3]

1 Ocean College, Zhejiang University, Zhoushan 316021, China; hotfireyao@163.com (Y.Y.); 21934102@zju.edu.cn (J.Z.); 22034207@zju.edu.cn (J.W.)

2 The Engineering Research Center of Oceanic Sensing Technology and Equipment, Ministry of Education, Zhoushan 316021, China

3 Hangzhou Xi'ao Environmental Technology Co., Ltd., Hangzhou 310005, China; yjx3051243007@163.com

* Correspondence: lilizju@zju.edu.cn

**Abstract:** Estuarine environmental capacity is the foundation for coastal biological diversity and self-purification capacity. Hence, studies on the marine environmental capacity (MEC) are the foundation for the total discharge control and water quality improvement of land-based pollutants. In the article, A calibrated two-dimensional hydrodynamic model was used to study the environmental characteristics of Sanmen Bay, including the tides, the residual currents, the tidal prism, and water exchange abilities. The model results were used to estimate the environmental capacity of the bay. Taking the pollution problem in Sanmen Bay as an example, the method of response factor, the sub-unit control method, and the phased control method were used to estimate the environmental capacity, pollutant amounts, and the pollutant reduction in the bay. The concentrations of COD, inorganic nitrogen, and acid salt in Sanmen Bay are spatially varied, with higher values occurring in the western part and in the inner bay. The half exchange time of the whole bay is about 23 days, and the exchange time of 95% water body is about 60 days. The evaluation of MEC cannot only provide technical support for the offshore aquaculture industries but also provide a scientific basis for the total control of terrigenous pollutants in coastal cities in Southern Zhejiang Province.

**Keywords:** marine environmental capacity (MEC); response factor; water exchange; residual; Sanmen Bay





## 1. Introduction

With the fast development of the coastal economy and marine aquaculture, estuaries are suffering a lot from terrestrial and oceanic pollutants. Understanding the characteristics of pollutants and estimating the marine environmental capacity (MEC) in estuaries provides a theoretical foundation for coastal management. The deteriorated environment of bays feeds back to and constrains the economy of coastal cities. Therefore, identifying sources and sinks of pollutants, and estimating the estuarine environmental capacity is quite important for coastal development (Yoon et al., 2020; Halpern et al., 2008; Syvitski et al., 2005) [1–3].

The main reason for the deterioration of the offshore water environment is that a large number of pollutants produced on land are discharged into it, including inorganic nitrogen, active phosphate, and heavy metals, etc. (Chen et al., 2008; Cui et al., 2013) [4,5].

The first step to conduct total discharge control is to assess the MEC of pollutants. (Wu et al., 2005) [6]. MEC refers to the maximum load of pollutants that can be accommodated in a specific sea area under the premise of making full use of the marine self-purification capacity without causing pollution damage (Linker et al., 2013) [7].

Sanmen Bay is located on the coast of the East China Sea (Figure 1a). It is a macro-tidal turbid estuary, with a maximum tidal range of approximately 2 m at the bay mouth and suspended sediment concentration (SSC) of 1.192 $kg/m^3$ at the middle of the bay. It has

four tributaries. With the fast development of human activities, the pollution problem is becoming increasingly heavy in the bay.

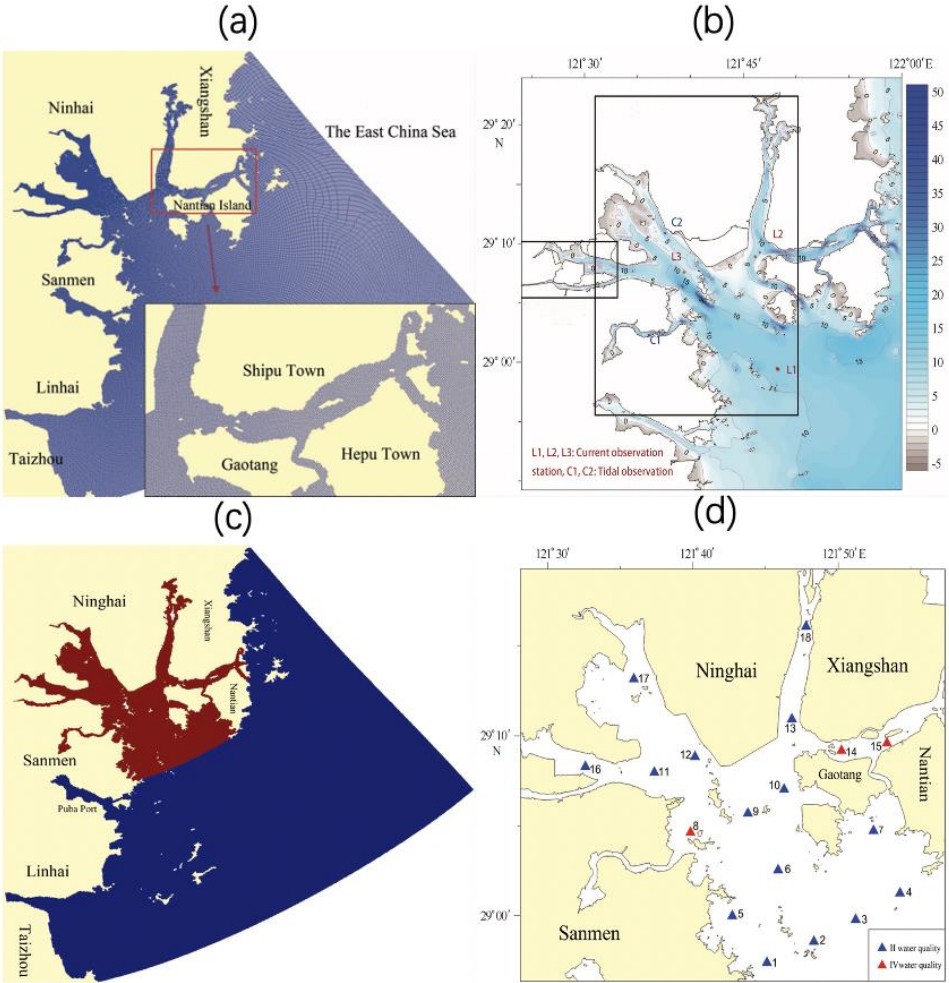

**Figure 1.** (**a**) Model domain for Sanmen Bay, (**b**) Location of observation stations, (**c**) Initial conditions of water exchange model (The concentration distribution in Sanmen Bay is set as 1 unit, red in the figure, the inflow outside the bay and at the boundary is set as 0, dark blue in the figure), (**d**) Water quality control stations in Sanmen Bay.

Identification of the contamination source is a part of the systematical analysis to find the pollution source fundamentally. Because how much a certain water body can hold is on the premise of pollution intensity which also plays the role to decide how to distribute for such a specific object.

According to the various forms of pollution being discharged into the water body, pollution sources can be divided into point source pollution and non-point source pollution. Non-point source pollution is referred to as water body pollution caused by rainfall run-off, this kind of contaminant entered into the soil or underground water body in a wide, microcrystalline, and dispersive way. Since the 1960s, research on non-point source pollution has caught the interest of scientists throughout the world. So far non-point source pollution feature, impact factor, load quantification on pollution output, and mechanism of pollutant migration and transformation have made great achievements.

A certain amount of contaminant emission poured is permitted. Because natural water body holds a certain amount of environmental capacity for a sort of pollution. Total emission radically depends on assimilative capacity, distribution of total water pollutants based on the limit of water environmental capacity.

According to the environmental quality survey results of the Sanmen Bay sea area from 2015 to 2016, and combined with the marine environmental function zoning, the researchers evaluated and classified the current situation of the environmental quality of the sea area (Liang et al., 2021) [8]. A 2-D hydrodynamic and pollutant model of Sanmen Bay is established based on Deift3D. Combined with the current hydrological status of Sanmen Bay, the transport and diffusion laws of COD, TP, and TN in the bay are analyzed (He et al., 2018) [9].

After conducting Clean Water Act in 1972, although industrial and municipal pollution has been under effective control by carrying out NPDES (National Pollutant Discharge Elimination System), the water quality has not been radically improved. Previous research showed that it is non-point source pollution that mainly polluted the rivers, lakes, and surface waters in estuaries. Also, non-point source pollution polluted underground water and degradation of wetland ecosystem. For such a reason, the Clean Water Act includes a guide rule, named TMDL (Total Maximum Daily Load), aiming at controlling both point source pollution and non-point source pollution. The acting emphasis is on figuring out non-point source pollution load and elimination in key water areas. Such a concept (MEC) was raised by Japanese scholars in the environment field in 1968. Japanese researcher Yanowokio (1968) claimed that environmental capacity is determined by environment quality standards, i.e., keeping total contamination loads within a permitted limit [10].

Streeter and Phelps (1952) raised a simple S-P model, which is the earliest form of water quality model [11]. The development of the water quality model can be concluded into 5 periods: 1925–1960, BOD-DO coupled model was raised based on the S-P model; during 1960–1965, spatial variety, physical, kinetic factors, and temperature were introduced into the model as a state variable, in the same the heat exchange between air and the water surface was also considered; during 1965–1970, as the computer started to be applied, people increasingly deeply understand biochemical oxygen consumption; the calculation method was developed from 1-D to 2-D; during 1970–1975, water quality model has developed into mutually non-linearity. In the last 20 years, space dimensionality has been into 3-D, the emphasis of study gradually turned into improving the dependability and evaluation of the model. In this way, dependability on the calculation of environmental capacity has been enhanced [12–16].

The eutrophication degree of seawater, the enrichment degree of heavy metals in sediments and the potential ecological hazard effects were comprehensively analyzed.

In this study, we take Sanmen Bay as an example, to study the characteristics of water exchangeability and marine environmental capacity, using both numerical models and field data.

Sanmen Bay is located in the monsoon subtropical humid climate zone. Affected by the monsoon climate, it has four distinct seasons and a mild climate. The weather changes are complex and disastrous weather is frequent. Disastrous weather of different degrees can be encountered in all seasons. The seasonal variation of temperature is obvious. Sanmen Bay has abundant rainfall, mainly from March to September. The whole year can be roughly divided into two rainy seasons and two relatively dry seasons. The wind direction of Sanmen Bay varies with seasons. Typhoons, rainstorms, and sudden small-scale disastrous weather occur from time to time.

## 2. Materials and Methods

### 2.1. Model Descriptions

A calibrated two-dimensional hydrodynamic model was used to reproduce the coastal oceanic and environmental characteristics of Sanmen Bay. Delft 3D simulates water surface elevation, velocity, water quality, waves, and morphology. Flow, Hydrodynamics were used in this study. The governing equations of the Delft3D hydrodynamic model in

the vertical sigma coordinate system and horizontal curvilinear coordinate system are expressed as follows:

$$\frac{\partial \zeta}{\partial t} + \frac{1}{\sqrt{G_{\xi\xi}}\sqrt{G_{\eta\eta}}}\frac{\partial\big((d+\zeta)U\sqrt{G_{\eta\eta}}\big)}{\partial\xi} + \frac{1}{\sqrt{G_{\xi\xi}}\sqrt{G_{\eta\eta}}}\frac{\partial\big((d+\zeta)V\sqrt{G_{\xi\xi}}\big)}{\partial\eta} = (d+\zeta)Q \quad (1)$$

where $\xi$ is the coordinate direction under the Delft3D curvilinear coordinate system corresponds to the X-axis of the rectangular coordinate system, $\eta$ is the Y-axis, $\zeta$ is the height of the water surface above the zero scale line of the Z coordinate, d is the depth from the zero scale line of the Z coordinate to the bottom of the water, $U$ is the velocity for X-axis, $V$ is the velocity for Y-axis, $\sqrt{G_{\eta\eta}}$ is conversion coefficient for X-axis and $\sqrt{G_{\xi\xi}}$ is conversion coefficient for Y-axis.

The three-dimensional convection-diffusion equation in the water quality module is as follows:

$$\frac{Cover\partial C}{\partial t} + v_x\frac{\partial C}{\partial x} - D_x\frac{\partial^2 C}{\partial x^2} + v_y\frac{\partial C}{\partial y} - D_y\frac{\partial^2 C}{\partial y^2} + v_z\frac{\partial C}{\partial z} - D_z\frac{\partial^2 C}{\partial z^2} = S + f_R(C,t) \quad (2)$$

where $C$ is substance concentration, $D$ is diffusion coefficient, $S$ is the inflow term, $f_R(C,t)$ is the reaction term.

## 2.2. Model Configurations

The model domain contains Sanmen Bay and its adjacent seas. The open ocean boundary is from 28°31′ N to 29°26′ N, the east part can extend to 122°27′ E. Jiaojiang River runoff is considered at the open boundary. The measurements were conducted in the bay between the Nantian station and the Linhai station (Figure 2a).

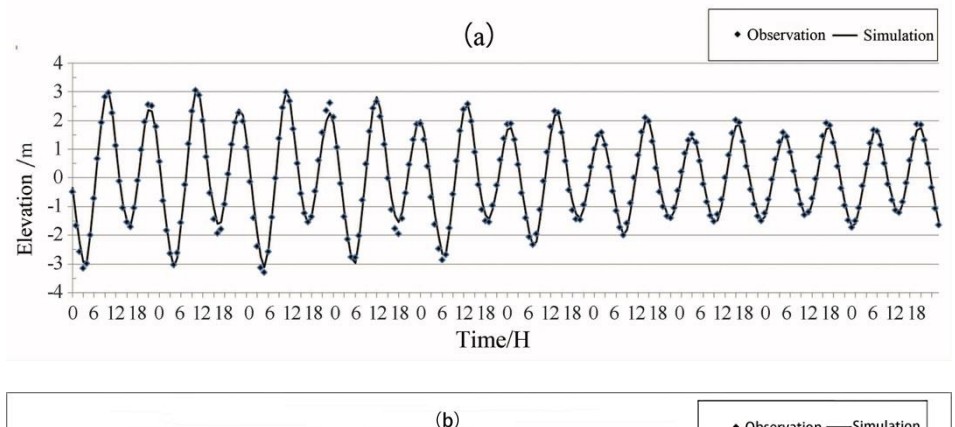

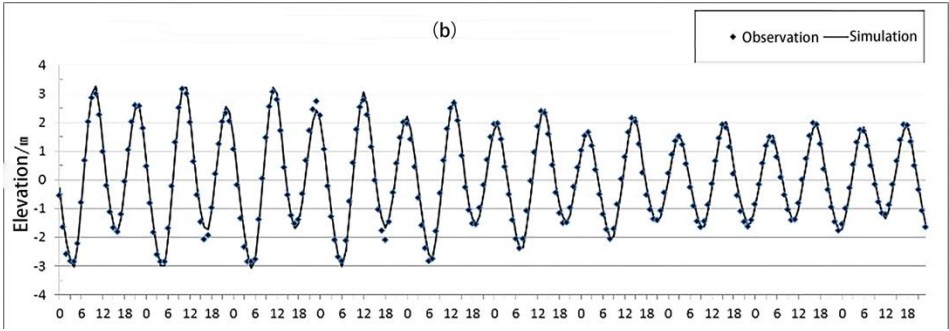

**Figure 2.** Tidal elevation in (**a**) Jiantiao station and (**b**) Gangdi station (2 December 2009 to 12 December 2009).

According to the calculation region, to generate orthogonal curvilinear grid scatter with the 2-D hydrodynamic model. The Mesh quantity is $629 \times 674$. In the study region, the minimal edge length is about 50 m, the maximum edge length is about 220 m, and the computational time step is set at 60 s. Charted depth data is gained from the historical chart. ADI method is used to solve the problem.

The computational domain is about 100 km on the x-axis and 110 km on the y-axis. The grid consisted of 423,946 elements, forming a mesh of orthogonal curvilinear grid with variable cell widths ranging from 220 m in the area of the open sea to 50 m in Sanmen Bay. The bathymetry data were interpolated linearly and were corrected with the satellite chart (Figure 1a).

The model was run for a further one month for the period from 1 December 2009 to 30 December 2009.

To calculate the environment capacity, we firstly determine water quality goal by dividing functional the water area, then conduct a numerical simulation to consider the quantitative response relation between pollution emissions and water environmental quality.

*2.3. Model Validation*

Hourly water elevation from 3 December to 12 December in 2009 is taken from Jiantiao station and Gangdi station are compared with model elevations in Figure 2. The result of verification basically agreed with the measured data. The relative error of the whole process could be controlled within 10%. According to spring tides and neap tides verification, the result can basically reflect the tide wave transformation of Sanmen Bay.

The current velocity and direction of each verification point coincides with the measured data, what's more, the current velocity summit and direction changing moment are both close to each other, and relative error could be controlled under 20%. In a conclusion, the numerical simulation in Sanmen Bay can basically reflect the regional hydrodynamic situation, which offers a basis for further research on water exchange and water environmental capacity (Figures 3 and 4).

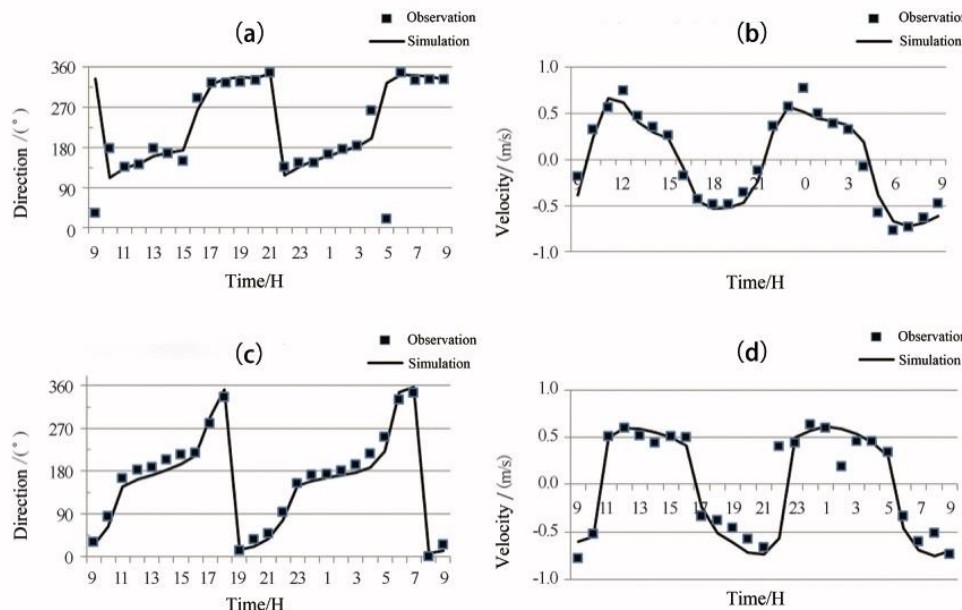

**Figure 3.** Current velocity and direction during spring tides (**a**,**c**) stand for direction, (**b**,**d**) stand for velocity.

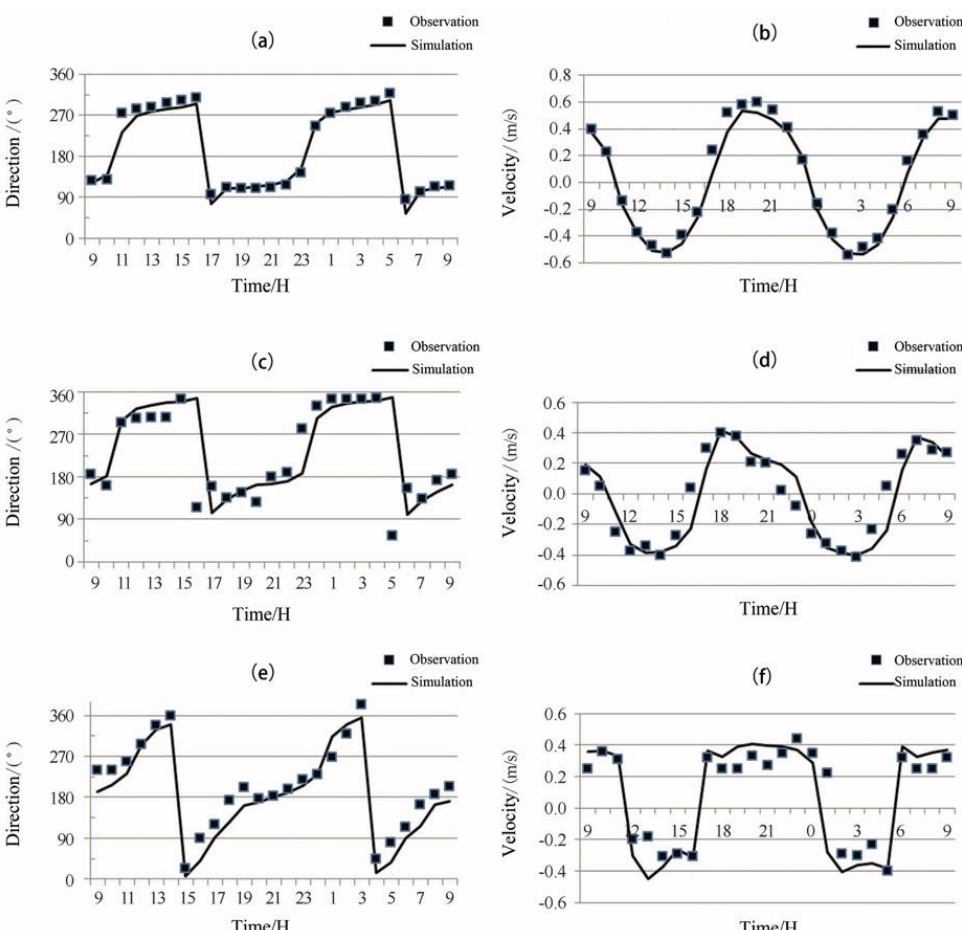

**Figure 4.** Current velocities and direction during neap tides, (**a**,**c**,**e**) stand for direction, (**b**,**d**,**f**) stand for velocity.

Compared with the measured values, the calculated values of the total tidal mean concentration in the whole sea area are also close, with an error of only 1.96%, indicating that the water quality model has successfully simulated the distribution of inorganic nitrogen concentration in Sanmen Bay.

The validation of COD is shown in Table 1. The model can simulate the contaminants well.

**Table 1.** Validation of COD.

| Station | Flood (Model) | Ebb (Model) | Average (Model) | Observation | Error/(%) |
|---------|---------------|-------------|-----------------|-------------|-----------|
| S1 | 0.6777 | 0.7055 | 0.6916 | 0.717 | −3.60 |
| S2 | 0.6225 | 0.6749 | 0.6487 | 0.661 | −1.91 |
| S3 | 0.5673 | 0.6360 | 0.6016 | 0.640 | −5.93 |
| S4 | 0.5399 | 0.5942 | 0.5671 | 0.556 | 2.06 |
| S5 | 0.5376 | 0.5483 | 0.5430 | 0.535 | 1.49 |
| S6 | 0.6963 | 0.7078 | 0.7020 | 0.728 | −3.51 |
| S7 | 0.6539 | 0.6997 | 0.6768 | 0.630 | 7.43 |
| S8 | 0.6105 | 0.6739 | 0.6422 | 0.583 | 10.25 |
| S9 | 0.6986 | 0.7020 | 0.7003 | 0.660 | 6.06 |
| S10 | 0.6865 | 0.6965 | 0.6915 | 0.594 | 16.51 |
| Average | 0.6432 | 0.6658 | 0.6545 | 0.642 | 1.96 |

*2.4. Method to Estimate the MEC*

The marine environment capacity (MEC) is decided by the following factors:

(1) The hydrogeology condition of the marine environment, such as marine space, locations, tidal conditions, self-purification abilities, and other natural conditions as well as group features of the ocean ecosystem.

(2) Rules on the utilization function of the specific maritime area usually differ. Different maritime functional areas carry out different water quality standards so that their water capacity varies.

(3) Physicochemical property. The environment capacity relies on self-purification which is decided by the physicochemical property of contaminants. Various kinds of contaminations do harm to people through different toxicities. Thus, environment capacity changes as the concentration permitted to exist differ.

The response coefficient method is adopted here for the research on environment capacity. By taking residual capacity maximum as principal, this study has Sanmen Bay for example, and then calculates the distribution of the marine environment capacity of Sanmen Bay.

Given that velocity and diffusion coefficient is determined, a convection-diffusion equation can be regarded as a linear equation.

The formula is shown as follows:

$$C(x,y,z) = \sum_{i=1}^{m} C_i(x,y,z) \tag{3}$$

where $C(x,y,z)$ means the concentration of each position (mg/L). $C_i(x,y,z)$ means the concentration of the $i$th position (mg/L).

The concentration field formed separately by each source can also be regarded as some times of the one formed when a certain source strongly discharges pollution.

$$C_i(x,y,z) = Q_i \cdot \alpha_i(x,y,z) \tag{4}$$

where $Q_i$ is the $i$th pollution source's emission; $\alpha_i$ is the response coefficient field of the $i$th pollution source, referring to one point's concentration under unit source intensity. $\alpha_i$ reflects the $i$th point's response level to the $i$th pollution source.

According to the response coefficient and controlling goals of each controlling point, to work out the environment capacity by the linear programming method. The main steps are as follows:

(1) After working out each source's response coefficient field, researchers should extract each controlling point's corresponding response coefficient value.

(2) Pollution source's permitted emission might correspond to the total residual emission maximum worked out by linear programming (LP) under the condition where controlling points meet the demand of water quality.

Taking Sanmen Bay, for example, this paper studies total amount control and emission cutdown management. One of the central basis of management of the marine environment is environment capacity. Contaminants picked to be calculated should reflect the water quality, degree of contamination, and operability on the management of environment capacity and contamination controlling and other aspects. Considering pollution source, the water quality of Sanmen Bay, and linking to the control index of land-sourced pollutants, Sanmen Bay's prime pollutants are nutrient salt formation such as nitrogen, phosphorus, etc. COD is a comprehensive index to describe the degree of water pollution. According to relevant provisions, CODs, (chemical oxygen demand) TN (total nitrogen), and TP (total phosphorus) should be taken into consideration in calculation of environment capacity or cutdown. Based on investigations of Sanmen Bay and relevant provisions on seawater quality, this paper will calculate by controlling the content of $COD_{Mn}$, inorganic nitrogen

and labile phosphate in seawater and then gets a result. Finally, the research will make a conversion from the result to environment capacity or cutdown for COD, TN, and TP.

## 3. Results

### 3.1. Hydrodynamics in the Bay

Field Observation and Analysis

　　Surface water elevations were measured by WSH at the Jiantiao station and Gangdi station, both recorded at a 60-min sampling interval. The velocity data discussed in this paper are measured by ADP at 400 KHz and 600 KHz in these two stations.

　　The tidal characteristic values of Jiantiao Station and Gangdi Station are less than 0.5, so it can be inferred that the tide of Sanmen Bay belongs to the regular semidiurnal tide (Table 2).

**Table 2.** Tidal eigenvalue ($H_{O_1}$, $H_{K_1}$ and $H_{M_2}$ means the amplitudes of $H_1$, $O_1$ and $M_2$ respectively).

| Date | $(H_{O_1}+H_{K_1})/H_{M_2}$ | |
|---|---|---|
| | **Jiaotiao Station** | **Gangdi Station** |
| 2009.7 | 0.36 | 0.32 |
| 2009.12 | 0.36 | 0.33 |

　　The current property ratios ($(W_{O1} + W_{K1})/W_{M2}$) of L1, L2, and L3 are all less than 0.5, indicating that the $M_2$ tidal component of Sanmen Bay is dominant and is the main tidal component. The tidal current type of Sanmen Bay belongs to the regular half day tidal current. Generally, the bay is greatly affected by the tidal component of the shallow sea. The calculation results also prove this point. Among them, the value ($W_{M4}/W_{M2}$) of theL1 station is the smallest. This is mainly because the L1 station is located at the mouth of Sanmen Bay, with an open water surface and deep depth, so it is less affected by the tidal component of the shallow sea (Table 3).

**Table 3.** Tidal current eigenvalue ($W_{O_1}$, $W_{K_1}$ and $W_{M_1}$ means the length of long semi axis of partial current ellipse of $O_1$, $K_1$ and $M_1$ respectively).

| Station | Layer | $(W_{O_1}+W_{K_1})/W_{M_2}$ | | $W_{M_4}/W_{M_2}$ | |
|---|---|---|---|---|---|
| | | **July 2009** | **December 2009** | **July 2009** | **December 2009** |
| L1 | surface | - | 0.283 | - | 0.059 |
| | 0.2H | - | 0.227 | - | 0.209 |
| | 0.6H | - | 0.244 | - | 0.049 |
| | 0.8H | - | 0.235 | - | 0.147 |
| | bottom | - | 0.267 | - | 0.133 |
| L2 | layer | 0.104 | 0.222 | 0.213 | 0.250 |
| | bottom | 0.061 | 0.292 | 0.182 | 0.208 |
| L3 | layer | 0.235 | 0.211 | 0.250 | 0.452 |
| | bottom | 0.206 | 0.348 | 0.267 | 0.688 |

### 3.2. Model Result Hydrodynamics in the Bay

　　Figure 5 shows the torrent velocity vector of ebb and flood tide during spring and neap tide in December 2009.

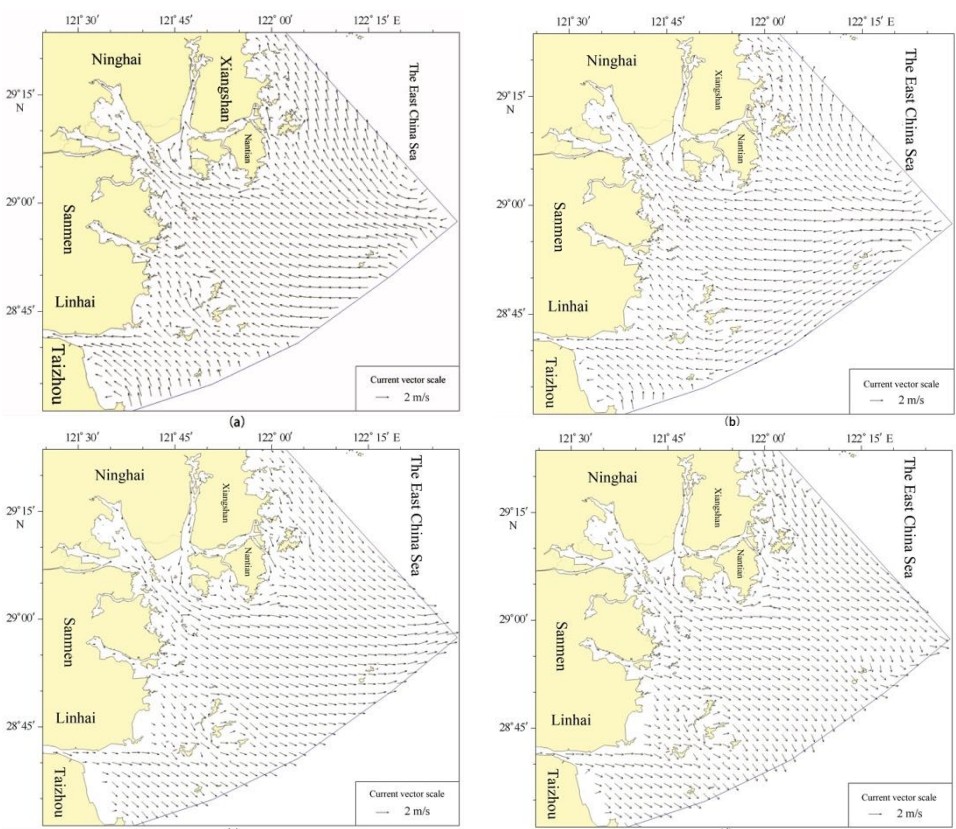

**Figure 5.** Depth-average velocity vectors during flood and neap (the entire domain). ((**a,b**) flood spring and flood ebb, (**c,d**) neap spring and neap ebb).

After the tidal waves propogate from the outer sea into Sanmen Bay through southeast to the northwest, the tidal motion characteristics are formed mainly with standing waves in the bay, while ebb and flood last little. The tidal current in Sanmen Bay is basically reciprocating, and the flow direction is mostly influenced by the topography. The flood tide flowing into the bay mouth is mainly in the northwest direction, and the ebb tide is mainly in the southeast direction. The water channels and harbor branches are basically along the longitudinal axis, among them Shipu Harbor's flood tide flow to the west, and ebb tide flow to the east. The open sea area from the bay mouth to the outer bay shows swirling current characteristics in different degrees.

During the flood period, the East China Sea tide waves propagate from the southeast side of the large area along the northwest direction, which mainly enters Sanmen Bay through its mouth. Most of the flood tide flows into the bay in the northwest through the bay mouth after passing through the Maotou Sea. A small part of the tide flows into the bay from east to west via Shipu Port. The two flood tides merge in the bay and then divide into four flows towards the summit of the bay. The first one flows into Jiantiao Harbor, the second one flows into Qimen Harbor and Haiyou Harbor through the Shepan water channel, the third one flows into Liyang Harbor and Qingshan Harbor, the fourth one flows into Baijiao water channel. The direction of the flood tide in each port branch is basically parallel to the longitudinal axis of the port branch, and the velocity is similar to that of the flood flow in the open sea. On the large area of shallow shoals which are apart from the port branch, the floodplain shows a slow flow and diffusing state.

During the ebb flow, the flow direction is basically opposite to that of flood flow, and the main stream is along southeast. The water at the bay summit leaks out of the trough. After the ebb tides converge, most of them flow out of the Sanmen Bay mouth in the southeast direction, and a small part of it flows out from west to east via Shipu Port. The Yushan Islands which is in the southeast corner of the calculation area and the Dongji

Islands in the southwestern part are both affected by the underwater topography, and the direction of the flood and ebb tides change to a certain extent, and the local currents are relatively complicated.

*3.3. Water Exchange Ability in the Bay*

Luff et al. (1996) introduced the concept of the Half-life period, which is defined as the time required for the concentration of the conservative substance to be diluted to half of the initial concentration by convection diffusion. The definition is based on the fact that it is almost impossible for the final concentration of a substance to be zero, and the rate of dilution represents the rate of water quality change, that is, the exchange capacity of the sea area.

Based on the concept of half exchange time, this study calculated the diffusion, transport, and dilution speed of conservative matter in each grid point of Sanmen Bay by using the transport and diffusion model of conservative materials, so as to study the water exchange capacity of Sanmen Bay.

3.3.1. Passive-Tracer Concentrations

Based on the previous hydrodynamic model, the water exchange model for the transport of regional passive-matter concentration is established (Figure 1c).

In the closed boundary condition, the current is 0 ($\frac{\partial C}{\partial n} = 0$), different treatment methods are adopted for the inflow and outflow of the model, and the inflow material concentration is 0, and the outflow material concentration is calculated by the model. The flow conditions are automatically obtained from the flow model.

According to the conservative material model, the whole continuous tidal process is applied as the calculation tide pattern, and the diffusion, transportation, and dilution process of the conservative material in the calculation water area are obtained continuously. Figure 6 shows the distribution of conservative materials at different times over three months. Note that the concentration value of conservative substances on each grid point not only represents its own concentration, but also is an important indicator of local water exchange degree. The period of the water exchange rate reaching 95% is about 60 days.

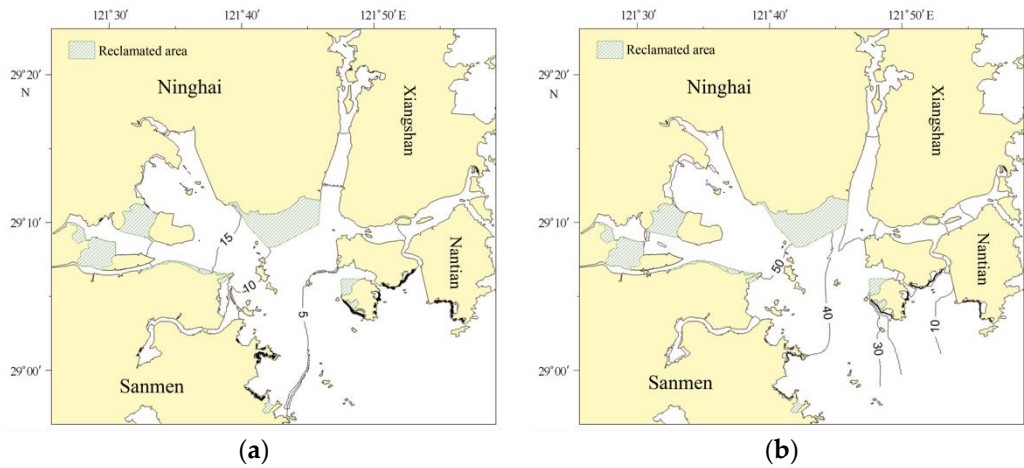

**Figure 6.** Water semi-exchange time (**a**) and 95% water exchange time (**b**).

After five days (Figure 7a), the water exchange degree of different regions in the bay is quite different. On the whole, the concentration of conservative material decreased gradually from the top of the bay to the estuary, indicating an increased exchange degree from the bay head to the bay mouth. At the same time, the water exchange degree of the bay mouth section increased gradually from the west to the east part of the bay. The direction of concentration isoline near the bay mouth was NNE-SSW, and the water exchange rate reached more than 90% on the fifth day in the bay mouth. The water exchange capacity of

Shipu port is relatively strong, and the water exchange rate from the bay head to the bay mouth rises from 40% to about 80%.

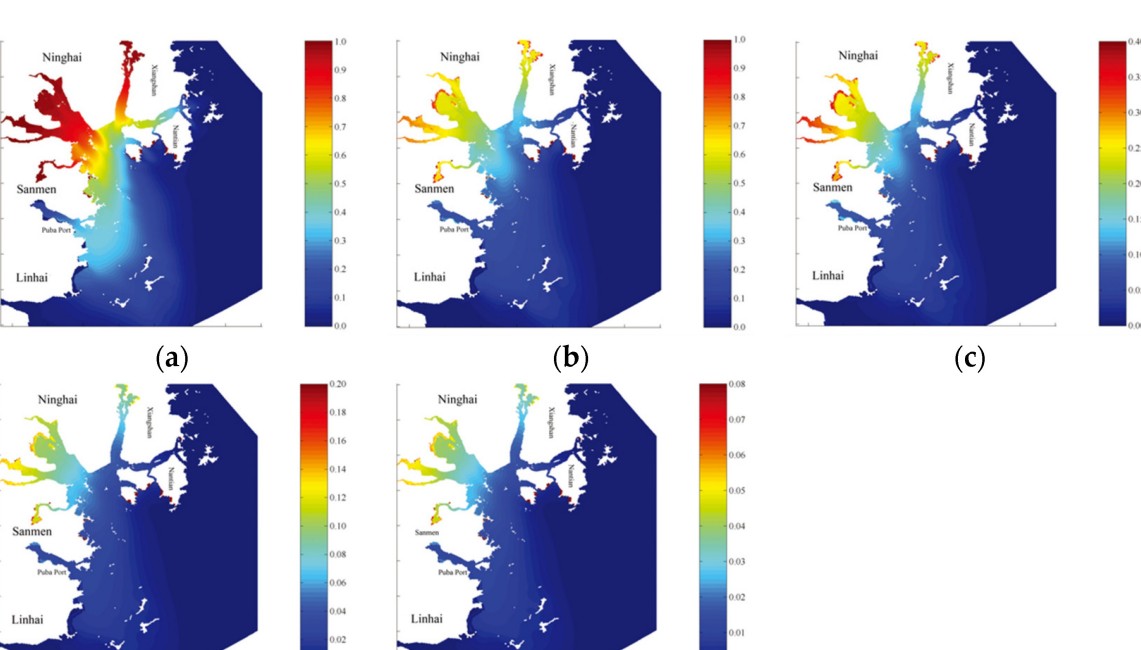

**Figure 7.** Conserved matter distribution after 5 days (**a**), 15 days (**b**), 30 days (**c**), 45 days (**d**) and 60 days (**e**).

Half a month later, most of the water areas in the bay that had not been exchanged five days ago have been exchanged to a certain extent, and the concentration value has dropped to less than 0.9 units.

One month later, except for the small tidal flats with high elevation in Xiaodao, most of the water bodies in Sanmen Bay have completed semi exchange, and the water exchange rate in the Bay has basically reached more than 60%. To facilitate the observation of the concentration distribution in the bay, the upper limit of concentration in the figure is changed to 0.4 units. The concentration isolines at the top and mouth of the bay continue to move into the bay, and the concentration gradient is significantly reduced. The overall distribution trend of the water exchange degree is still similar to that before. The concentration contour line in the western part of the bay is roughly along the NE-SW direction, and the relatively high concentration waters are mainly located in Qimen port and Haiyou harbor, with a concentration value of between 0.25 and 0.35. The concentration of Shipu port in the bay head has dropped below 0.1 unit.

Figure 8d,e shows that the trend of concentration distribution and isoline trend in the bay are close, while the overall concentration decreases and the concentration contour continues to be extrapolated. After one and a half months, the exchange ratio of all water bodies in the bay reaches more than 85%; after two months, the exchange ratio of most water bodies in the bay reaches more than 90%, except for some waters at the top of the west of the Bay. It can be considered that the water exchange in Sanmen Bay has been basically completed at this time.

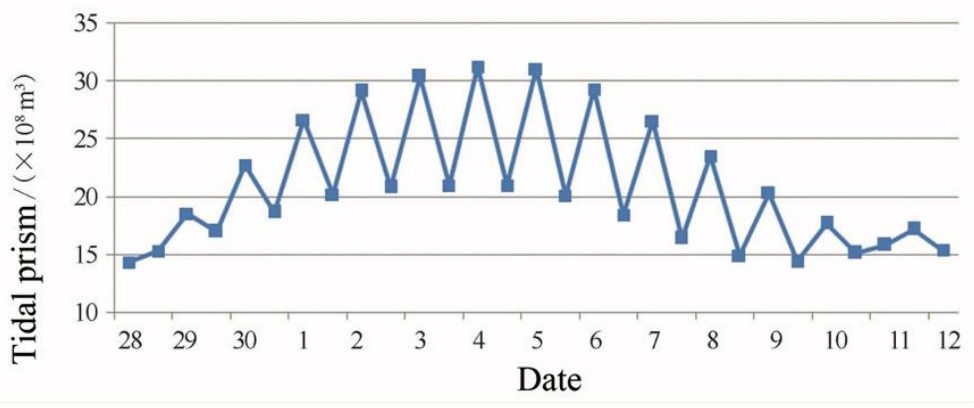

**Figure 8.** Tidal prism changing process (28 November 2009 to 12 December 2009).

3.3.2. Water Exchange Ability

Tidal prism means the volume of tidal water that a certain bay can hold. It is an important indicator of the environmental assessment of a bay and an important parameter reflecting the exchange of sea water in the bay. The amount of tidal prism is of great significance to the marine environment, the exchange of water bodies in the bay, the maintenance of the port branch, and the water depth in the channel. Sanmen Bay is a typical semi-enclosed bay, the rivers flowing into the bay is mountainous rivers. Therefore, when calculating the tidal volume of Sanmnen Bay, the tidal volume flowing out of the coastal estuary from the calculation area is ignored, and only the part in Sanmen Bay is considered. Two tidal current channel sections between the bay and the open sea, the Sanmen Bay section and Shipu Port are perpendicular to the longitudinal axis of the main waterway and Shipu section (Figure 8).

The tidal volume is defined as the newly added tidal volume entering the bay from the low tide time to the high tide time in any one tide cycle. The tidal volume of a tidal cycle can be expressed as:

$$Q = \int_{T_{low}}^{T_{high}} \int_{A_1} U_1 D_1 dA_1 dt + \int_{T_{low}}^{T_{high}} \int_{A_2} U_2 D_2 dA_2 dt \tag{5}$$

The hydrodynamic model is ideal for the simulation results of regional tides and tidal current processes. The simulated flow field can basically reflect the calculation of regional hydrodynamics. The calculation results can be used as the basis for the study of water exchange and water environment capacity in Sanmen Bay.

The tidal volume of Sanmen Bay is relatively large. It's between $15 \times 10^8 \sim 30 \times 10^8 \mathrm{m}^3$ during a spring-neap tidal cycle, $20 \times 10^8 \sim 30 \times 10^8 \mathrm{m}^3$ during spring tides, and $15 \times 10^8 \sim 18 \times 10^8 \mathrm{m}^3$ during neap tides. The average tidal prism is about $20.78 \times 10^8 \mathrm{m}^3$.

According to the results of numerical calculation of water exchange, the distribution of semi exchange capacity in Sanmen Bay varies greatly in different regions of the bay. Generally speaking, the water exchange capacity of the bay mouth and Shipu port is strong, and the water exchange in the west of the bay is relatively slow compared with that in the East. On the whole, the half exchange time of water in Sanmen Bay is less than 23 days, and 95% of the water exchange period is within 50 days in relatively open main water area; the half exchange time of water body in most areas of the branches is more than 10 days, and 95% of the water exchange period is more than 50 days; the water exchange time of the west end of Shipu port is longer than that of the east end, the half exchange period is less than 8 days, and 95% of the water exchange period is less than 40 days.

## 4. Discussion

### 4.1. Estimation of Environmental Capacity

Based on the precondition that to maintain the function of the environmental capacity of water, pollutant emission maximum that receiving waterbody can endure, that is to say, are permissible quantity of pollutants under the goal of water quality and hydrological condition. According to the situation of the water environment in Sanmen Bay, to calculate environment capacity about $COD_{Mn}$ is to figure out the maximum value of emissions' sum from pollution loads at all outfalls. Gaining value which is as great as possible on environment capacity is important. Yet important management methods for contaminants controlling and environment protection might coordinate with the operability of management.

Working out a linear programming problem is to figure out a linear function's maximum or minimum under the constraint of a set of linear equations or inequations.

The response coefficient method for environment capacity calculation transforms into a linear programming maximization problem [12]. It means:

Object function:

$$max \sum_{j=1}^{n} Q_j \tag{6}$$

Constraint equation:

$$C_{0i} + \sum_{j=1}^{n} \alpha_{ij} Q_j \leq C_{si}, (i = 1, 2, \ldots, m) \tag{7}$$
$$Q_j \geq 0, (j = 1, 2, \ldots, n)$$

where, $j$ means pollutant sources' serial number, $n$ means the number of sources; $i$ means the serial number of control points of water quality, $m$ means the number of control points of water quality; $C_{0i}$ means background concentration of control points; $\alpha_{ai}$ means the coefficient of the $j$th pollution source's emission at the $i$th control points of water quality.

To quickly obtain the maximum value, the problem was transformed into the standard forms in linear programming. Let $C_i = C_{si} - C_{0i}$, to convert the inequation into an equivalent equation.

Object function:

$$maxQ = \sum_{j=1}^{n} Q_j \tag{8}$$

Constraint equation:

$$\sum_{j=1}^{n} \alpha_{ij} Q_j \leq C_i, (i = 1, 2, \ldots, m) \tag{9}$$

$$Q_j \geq 0, (j = 1, 2, \ldots, n) \tag{10}$$

where, $C_i$ means the $i$th concentration capacity of the control points of water quality. Environment capacity assessment in this study involved COD, labile phosphate, and inorganic nitrogen. Control stations distribution can be seen in Figure 1d.

#### 4.1.1. The Responses Factor Field of COD

According to the method of response factor, the response factor field of pollution source in each catchment unit needs to be calculated first. The response factor field uses the same region and mesh as the pollution diffusion model. The initial condition was set as 0 to exclude other source strength's influence. The concentration field (also known as response factor field) is as Table 4 shows.

**Table 4.** $COD_{Mn}$ concentration field (mg/L).

| Number | Bapu | Jiantiao | Haiyou | Qimen | Liyang | Baijiao | Shipu |
|---|---|---|---|---|---|---|---|
| 1 | 0.0043 | 0.0035 | 0.0030 | 0.0028 | 0.0030 | 0.0019 | 0.0010 |
| 2 | 0.0011 | 0.0026 | 0.0025 | 0.0023 | 0.0025 | 0.0017 | 0.0010 |
| 3 | 0.0005 | 0.0015 | 0.0015 | 0.0015 | 0.0016 | 0.0012 | 0.0010 |
| 4 | 0.0002 | 0.0004 | 0.0005 | 0.0005 | 0.0005 | 0.0008 | 0.0012 |
| 5 | 0.0039 | 0.0046 | 0.0037 | 0.0037 | 0.0036 | 0.0021 | 0.0010 |
| 6 | 0.0010 | 0.0042 | 0.0053 | 0.0053 | 0.0054 | 0.0032 | 0.0013 |
| 7 | 0.0003 | 0.0010 | 0.0012 | 0.0012 | 0.0013 | 0.0034 | 0.0037 |
| 8 | 0.0010 | 0.0115 | 0.0097 | 0.0097 | 0.0096 | 0.0028 | 0.0012 |
| 9 | 0.0008 | 0.0040 | 0.0059 | 0.0059 | 0.0064 | 0.0053 | 0.0019 |
| 10 | 0.0007 | 0.0025 | 0.0029 | 0.0029 | 0.0029 | 0.0157 | 0.0039 |
| 11 | 0.0008 | 0.0057 | 0.0239 | 0.0239 | 0.0150 | 0.0029 | 0.0012 |
| 12 | 0.0008 | 0.0052 | 0.0108 | 0.0108 | 0.0196 | 0.0037 | 0.0014 |
| 13 | 0.0005 | 0.0020 | 0.0024 | 0.0024 | 0.0025 | 0.0281 | 0.0039 |
| 14 | 0.0005 | 0.0020 | 0.0023 | 0.0023 | 0.0024 | 0.0114 | 0.0069 |
| 15 | 0.0004 | 0.0016 | 0.0019 | 0.0019 | 0.0020 | 0.0077 | 0.0076 |
| 16 | 0.0007 | 0.0052 | 0.0273 | 0.0273 | 0.0140 | 0.0027 | 0.0011 |
| 17 | 0.0007 | 0.0051 | 0.0118 | 0.0118 | 0.0198 | 0.0028 | 0.0011 |
| 18 | 0.0004 | 0.0019 | 0.0022 | 0.0022 | 0.0023 | 0.0380 | 0.0035 |

### 4.1.2. Environmental Capacity of $COD_{Mn}$

To figure out $COD_{Mn}$ permitted emission at each catchment according to Linear Programming, this study took the maximum concentration during a whole tide as background concentration to estimate the capacity. Parameters about the permitted emission of $COD_{Mn}$ are shown in Table 5.

**Table 5.** Parameter on $COD_{Mn}$ (mg/L).

| Number | $C_{0i}$ | $C_{si}$ | Number | $C_{0i}$ | $C_{si}$ |
|---|---|---|---|---|---|
| 1 | 0.952 | 3 | 10 | 0.827 | 3 |
| 2 | 0.944 | 3 | 11 | 0.852 | 3 |
| 3 | 0.921 | 3 | 12 | 0.844 | 3 |
| 4 | 0.882 | 3 | 13 | 0.815 | 3 |
| 5 | 0.892 | 3 | 14 | 0.820 | 5 |
| 6 | 0.857 | 3 | 15 | 0.830 | 5 |
| 7 | 0.852 | 3 | 16 | 0.877 | 3 |
| 8 | 0.842 | 5 | 17 | 0.849 | 3 |
| 9 | 0.837 | 3 | 18 | 0.811 | 3 |

Linear programming solving was conducted on 7 catchment units based on the maximum remaining capacity. The maximum remaining emission permitted can be seen in Table 6. $COD_{Mn}$ concentration of each control station and utilization rate can be seen in Table 7. Based on the result calculated with present source strength, concentration distribution was used to estimate the emission permitted which shows that only Puba and Shipu can still discharge. The result of solving the constraint conditions of each control station is concentrated in two stations, because the calculation of linear programming is carried out according to the mathematical conditions. When selecting the largest group of total capacity among all feasible solution, it is obviously reasonable for the catchment which is relatively close to the open sea.

**Table 6.** Emission permitted of COD ($\times 10^4$ $t/a$).

| Catchment | Puba | Jiantiao | Haiyou | Qimen | Liyang | Baijiao | Shipu |
|---|---|---|---|---|---|---|---|
| COD Emission Permitted | 13.34 | 0 | 0 | 0 | 0 | 0 | 18.22 |
| Overall Amount | | | | 31.56 | | | |

**Table 7.** Concentration of each control station concentration at the maximum environmental capacity of COD.

| Control Station Number | COD Concentration | Utilization Rate | Control Station Number | COD Concentration | Utilization Rate |
|---|---|---|---|---|---|
| 1 | 3.000 | 100.00 | 10 | 3.000 | 100.00 |
| 2 | 1.827 | 60.90 | 11 | 1.760 | 58.67 |
| 3 | 1.572 | 52.40 | 12 | 1.850 | 61.67 |
| 4 | 1.526 | 50.87 | 13 | 2.934 | 97.80 |
| 5 | 2.819 | 93.97 | 14 | 4.444 | 88.88 |
| 6 | 1.874 | 62.47 | 15 | 4.769 | 95.38 |
| 7 | 2.812 | 93.73 | 16 | 1.680 | 56.00 |
| 8 | 1.789 | 35.78 | 17 | 1.657 | 55.23 |
| 9 | 2.088 | 69.60 | 18 | 2.721 | 90.70 |

Table 7 shows Station 1 and Station 10 have reached the limited values. According to the prediction results of COD environmental capacity, without considering the uniformity principle, only from the aspect of maximum source strength increment, under the condition of meeting the control objectives, the maximum COD pollutant discharge capacity of Sanmen Bay is about $31.56 \times 10^4 t/a$, which is the maximum theoretical calculation result under the calculation conditions.

*4.2. Reduction of Main Pollutants*

As the nitrogen and phosphorus nutrients in Sanmen Bay have exceeded the standard, different emission reduction schemes for inorganic nitrogen and active phosphate should be analyzed. In this paper, the phased control method is used to calculate the pollutant reduction. Firstly, pre-calculation is conducted to analyze the influence of the variation of pollution source strength of each catchment on the distribution of the concentration field, so as to provide the basis for determining the formal calculation scheme and preliminarily determine the minimum reduction required to reach the target. Secondly, the reduction scheme is determined according to the phased control target. Finally, the results of each scheme were compared and selected, and the reduction of inorganic nitrogen pollutants in each catchment was determined on the basis of meeting the requirements of the phased control index for environmental capacity calculation of Sanmen Bay.

The water quality model was used to simulate a different reduction of inorganic nitrogen source strength, and the concentration of sea area under 0.65 mg/L was found to be relevant to the reduction of source strength [17,18]:

$$S = 336.43e^{0.9146x} \tag{11}$$

$S$ means area, $x$ means percentage of source strength.

According to the model result, when the recent source strength reduction reaches 14.0%, the available sea area reaches 378.16 km$^2$, which accounts for 60.28% of the total area in Sanmen Bay. Mid-term and long-term is long from the present, the natural conditions and socio-economic conditions may change significantly so that the prediction is likely to deviate (Figure 9). The water quality improvement target needn't be completed accurately. According to the model result, when the source strength reduction reached 30.0% and 44.0%, the available area can reach 441.72 km$^2$ in middle term and 497.62 km$^2$ in long term (Table 8).

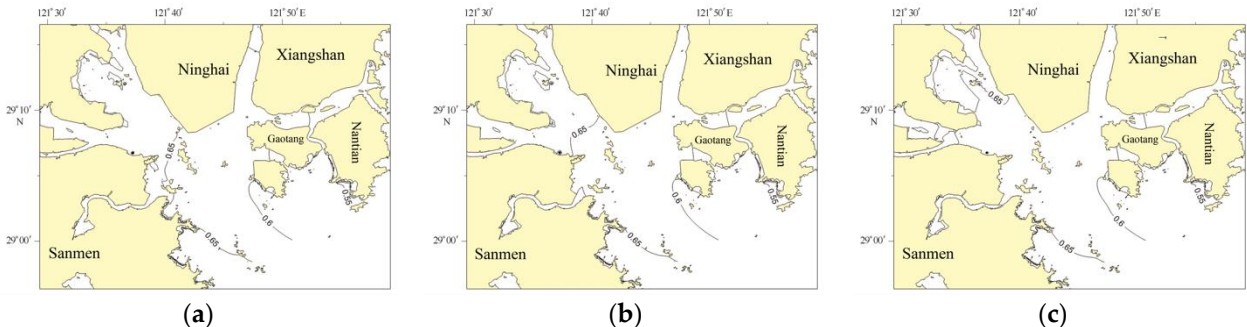

**Figure 9.** Result of source strength reduction in near term (**a**) mid-term (**b**) and long term (**c**).

**Table 8.** Result of source strength reduction.

| Number | Reduction Rate of Source Strength/% | Area/km |
|--------|-------------------------------------|---------|
| 1 | 5 | 356.08 |
| 2 | 10 | 369.59 |
| 3 | 15 | 384.04 |
| 4 | 20 | 399.52 |
| 5 | 25 | 421.77 |
| 6 | 30 | 445.04 |
| 7 | 35 | 461.04 |
| 8 | 40 | 480.69 |
| 9 | 45 | 510.17 |
| 10 | 50 | 536.15 |

## 5. Conclusions

A calibrated two-dimensional hydrodynamic model was built and fully validated to study the environmental characteristics of Sanmen Bay, including the tides, the residual currents, the tidal prism, and water exchange abilities.

Tides in the bay are regular semidiurnal tides, and the average tidal range is more than 4 m. The shallow water component has a certain influence on the tidal currents. The SSC in the bay is high, and is mainly caused by tidal current. The average tidal prism of the bay is about $20.78 \times 10^8 \mathrm{m}^3$.

The distribution of semi-exchange capacity of water bodies varies greatly in different regions of the bay. Generally speaking, the water exchange capacity of the bay mouth and Shipu port is strong, and the water exchange in the west of the bay is slower than that in the East. The half exchange time of the whole bay is about 23 days, and the exchange time of 95% water body is about 60 days; the half-exchange time of relatively open sea area is less than 15 days, and 95% of water exchange time is about 50 days.

The concentrations of COD, inorganic nitrogen, and acid salt in Sanmen Bay showed a trend of being higher in the inner estuary and lower outside of the bay, and was higher in the western part and lower in the eastern part. The concentration of COD was lower than 0.60 mg/L in most areas of the eastern part of the bay, while was higher than 0.65 mg/L in the western part of the bay. The concentration of inorganic nitrogen was more than 0.70 mg/L near the west coast. The concentration of acid salt was lower in the outer bay, while was higher in the inner bay.

**Author Contributions:** Methodology, Y.Y. and J.Y.; software, J.Z.; validation, J.Z.; investigation, Y.Y.; resources, Y.Y.; writing—original draft preparation, J.Z.; visualization, J.Z.; supervision, L.L. and J.W. All authors have read and agreed to the published version of the manuscript.

**Funding:** This research was funded by the Science Technology Department of Zhejiang Province (2020C03012, 2022C03044).

**Institutional Review Board Statement:** Not applicable.

**Informed Consent Statement:** Not applicable.

**Data Availability Statement:** Not applicable.

**Acknowledgments:** This research was partially supported by a grant from the Science Technology Department of Zhejiang Province. Data were provided by Qin Chen.

**Conflicts of Interest:** The authors declare no conflict of interest. The funders had no role in the design of the study; in the collection, analyses, or interpretation of data; in the writing of the manuscript, or in the decision to publish the results.

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
