# Peer review of "Marine Environmental Capacity in Sanmen Bay, China"

_water, doi:10.3390/w14132083_

Round 1
Reviewer 2 Report
The work is well prepared. However, it requires minor changes in scope:
1. correction of the aim of the work to be more interesting and taking into account the verification of the research hypothesis
2. changing the position of Fig. 1
3. correcting the position of figures so that they are closest to being cited
4. increasing the readability of figures 2-4
5. correcting chapter titles. I do not understand the notation "analyses" instead of "results".
Author Response
- I don't definitely understand the first comment, if you mean the hypothesis, when we calculated, In order to eliminate the influence of other source intensities on the concentration field formed by pollutant source intensities in each catchment unit, the boundary conditions and initial conditions are all taken as 0. When calculating the response coefficient field of a catchment unit, the pollutant emission of the catchment unit is taken as 1t/d, and the pollutant source intensity of other catchment units is taken as 0 to calculate the pollutant diffusion. And tides, currents and contaminants have been verified.
- Its position has been changed.
- Figures' positions have been checked, words revised is shown in blue.
- Some words have been added to explain fig.2-fig.4. I hope they can add readability.(Line 165 to Line 179)
- 'Results' has taken place of 'Analysis' .
